# Embodied Third-Person Virtual Locomotion using a Single Depth Camera

Aniruddha Prithul*  Isayas Berhe Adhanom †  Eelke Folmer‡

Department of Computer Science and Engineering
University of Nevada, Reno

## ABSTRACT

Third-person is a popular perspective for video games, but virtual reality (VR) seems to be primarily experienced from a first-person point of view (POV). While a first-person POV generally offers the highest presence; a third-person POV allows users to see their avatar; which allows for a better bond, and the higher vantage point generally increases spatial awareness and navigation. Third-person locomotion is generally implemented using a controller or keyboard, with users often sitting down; an approach that is considered to offer a low presence and embodiment. We present a novel third-person locomotion method that enables a high avatar embodiment by integrating skeletal tracking with head-tilt based input to enable omnidirectional navigation beyond the confines of available tracking space. By interpreting movement relative to an avatar, the user will always keep facing the camera which optimizes skeletal tracking and keeps required instrumentation minimal (1 depth camera). A user study compares the performance, usability, VR sickness incidence and avatar embodiment of our method to using a controller for a navigation task that involves interacting with objects. Though a controller offers a higher performance and usability, our locomotion method offered a significantly higher avatar embodiment.

**Index Terms:** I.3.7 [Computer Graphics]: Three-Dimensional Graphics and Realism—Virtual Reality

## 1 INTRODUCTION

One unique feature of virtual reality (VR) is that it can let you experience being a person of a different race, gender or age. Embodiment illusion research explores creating illusions of ownership over a virtual body, which is a promising intervention technique to reduce biases. For example, a seminal study [37] demonstrated that when light skinned participants experienced being a dark-skinned avatar (for example: Figure 1) this reduced implicit racial bias against dark-skinned people. A more recent study [48] explored swapping genders and found this to lead to reduced gender-stereotypical beliefs. These findings demonstrate the vast potential of VR as a tool to improve the world that could address many current day social issues regarding race, age and gender. Because VR is predominantly experienced from a first-person-perspective (1PP), to establish the embodiment illusion, a virtual mirror is required [17] which allows subjects to fully see themselves. Requiring a stationary mirror to maintain the embodiment illusion, limits what kind of scenarios can be explored. It might be interesting to explore non-stationary scenarios (i.e., are you treated differently based on skin color, gender, or age when walking through a busy street?). But that would require the ability for the user to move around in a virtual environment while still being able to see themselves.

Presence, i.e., a sense of being in VR [29], and embodiment, i.e., a sense of being your avatar [26] are important yet closely intertwined

---

*e-mail: aprithul@gmail.com
†e-mail: iadhanom@nevada.unr.edu
‡e-mail:eelke@unr.edu

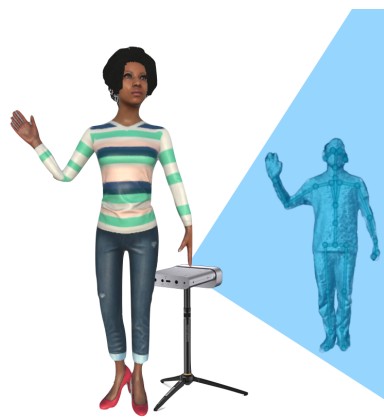

Figure 1: Our 3PP locomotion method integrates full-body skeletal tracking using a single depth camera (Azure Kinect) with head-tilt based input using inertial as to enable omnidirectional navigation beyond the confines of available tracking space. With the depth camera at the location of the avatar and interpreting movement relative to the avatar, the user always faces the camera which optimizes full-body tracking and keeps instrumentation to a minimum.

qualities for VR that to a large extent are defined by the graphical perspective used [18]. With 1PP users see the world through the eyes of their avatar -which provides the highest sense that they are their avatar [39, 46] (i.e., embodiment). Because the user and avatar are collocated, this view is most natural to us, and most optimal for motor accuracy tasks [32, 44].

A third-person perspective (3PP) allows users to see their avatar from an over-the-shoulder view. Though a 3PP offers a lower embodiment than 1PP [11]; when users can see their avatar it creates a stronger bond than when using 1PP and the ability to customize the avatar can create a strong identity which reinforces body ownership [12]. The higher vantage point improves spatial awareness [41, 45], though avatar occlusion can interfere with precise interaction like aiming [44]. Letting users switch between 1PP and 3PP offers the benefits of each perspective [43]. Because the user and their avatar are not collocated, 3PP generally offers a lower embodiment than 1PP [46]. This can be increased through agency, i.e., when the user and virtual avatar are linked to move synchronously [27].

Locomotion is an essential part of VR interaction [58] and defines presence [8]. Full body tracking enables visuo-motor synchronicity between the user and the avatar and when combined with real walking it achieves a high presence and sense of embodiment [18, 32]. Though real walking offers the highest presence [53], it is bounded by available tracking and physical space [7]. To implement omnidirectional 3PP locomotion, full-body tracking requires multiple cameras [32] or extensive user instrumentation [18]. Current consumer VR systems do not feature full-body tracking and positional tracking is limited to tracking the head mounted display (HMD) and both controllers. To implement 1PP, limited tracking is not an issue as users only see their hands, but for 3PP, tracking only 3 joints is not sufficient for animating an avatar that offers high embodiment [18].

Because users need to be able to navigate beyond the confines of available walking space, 3PP locomotion is typically implemented with a controller. A controller offers low presence and embodiment which is further exacerbated by the fact that users will often sit down when positional tracking is not fully used [60]. Currently few VR experiences use 3PP, though it is a popular perspective for non-VR games (e.g., Fortnite). Though some have suggested that 3PP is not a suitable perspective for VR [32], there are benefits of using 3PPs that have not fully been investigated yet. Locomotion facilitated using a controller (i.e., joystick or touchpad) is more likely to induce VR sickness [23,53]; a major barrier to the success of VR. Especially sudden movements, e.g., jumping, falling or users taking damage without corresponding head motions can exacerbate visual-vestibular conflict; a major cause of VR sickness [34]. One remedy is to use a rest frame [40], i.e., part of the screen with no optical flow, for example, a virtual nose [56]. Visual discomfort can also result from a vergence-accommodation conflict (VAC) [22]. VR headsets use a flat screen to simulate depth of field, which creates a disparity between the focal point of objects in the virtual world (vergence) and the actual physical surface of the screen (accommodation).

For 3PP, because the camera follows the avatar from behind, it can largely dampen out sudden motions [49] like a steady-cam. Because the avatar is likely the focus of a user's gaze during locomotion it serves as a rest frame. When using 1PP users are looking at objects in their entire field of view, but the higher vantage point of 3PP basically confines objects to a plane in the lower half of the users' view, which might alleviate VAC [50]. Developing a better understanding how perspective affects visual discomfort and VR sickness is important for VR [5].

We present a hybrid 3PP locomotion method that offers a high sense of embodiment by integrating real walking -implemented using full-body tracking- with tilt-based omnidirectional locomotion. Our goal was to bring a popular gaming perspective to the domain of VR while preserving a high embodiment. Our interface could enable embodiment illusion research [37] from a 3PP, and thus would not require users to look at themselves in a mirror using 1PP. It advances over existing 3PP methods [18,32] as locomotion isn't confined by available tracking space and removes the need to hold a non immersive controller [13]. Due to the unique implementation of steering, the user always faces the camera and our approach is minimal in terms of required instrumentation. In addition to understanding how perspective affects embodiment and motor accuracy, we investigate VR sickness.

## 2 RELATED WORK

A number of studies have investigated how perspective affects presence, motor accuracy and embodiment [26] for HMD based VR. Factors that can affect embodiment include: location of the body, body ownership, agency and motor control; and external appearance [16].

Salamin et al. [43] was one of the first studies to investigate perspective for HMD based VR and found some preliminary evidence that 3PP is better for navigation while motor actions like opening a door or putting a ball in a cup had better performance in 1PP. A follow up study by the same authors [42] found no difference in error rate between perspectives for a stationary ball catching task though a 3PP offered better distance estimation. Slater et al. [46] evaluated how perspective affects the body transfer illusion and found 1PP offered the highest embodiment. Debarba et al. [10] conducted an extensive study investigating perspective and synchronous/asynchronous avatar rendering on the performance for a target reaching task. Full-body motion capturing was implemented using an optical tracking system with wearable markers. Synchronous rendering of an avatar offered the highest performance and embodiment in terms of body ownership and self location with no difference between perspectives. A follow up study by the same

authors [15] evaluated perspective for a similar target reaching task and found that giving users the option to switch between 1PP and 3PP offers a strong sense of embodiment, though subjective body ownership was strongest in 1PP. Gorisse et al. [18] evaluates perspective on performance, presence and embodiment for an object perception and deflection task as well as a navigation and interaction task. No difference between perspectives on presence or agency were found though a 1PP enables more accurate interactions with objects while a 3PP provides better spatial awareness. Medeiros et al. [32] performed an extensive study that investigated perspective and avatar realism on navigation performance and embodiment. Navigation tasks using real walking included avoiding objects and going through a tunnel. Full body motion capture was implemented using an array of 3D depth cameras (Kinect). A 3PP offered the same sense of embodiment, spatial awareness and navigation performance as a 1PP when using a realistic representation, but did worse without realism.

Focusing on 3PP locomotion methods that enable navigation at scale, the following approaches are closely related. Hamalainen et al. [21] presents Kick Ass Kung-Fu; a martial arts installation that captures the user with a regular camera and embeds their graphics and translates their movements to an avatar in a 2D fighting game. Oshita [35] presents a motion capture framework for 3PP locomotion for large screen VR. Full body motion capture is implemented using an optical tracking system and a combination of walking-in-place and arm swinging allows for navigating beyond tracking space confines. Omnidirectional navigation is not supported due to the requirement to keep facing the screen. No user studies results were reported. Work by the same author [36] explored using hand gestures to control an avatar in 3PP on a large screen. Locomotion is achieved using walking in place with the fingers controlled by the user's right hand. A user study demonstrated the feasibility of this approach but revealed issues with locomotion at scale.

Cmentowski et al. [9] presents a VR locomotion method called Outstanding that lets users switch between 1PP and 3PP. Other than physical walking using positional tracking there is limited travel in 1PP and users switch to 3PP to travel beyond the limits of available tracking space. The camera remains stationary to avoid optical flow generation that could lead to VR sickness. When in 3PP, users navigate their avatar by pointing at a destination using a raycast with their motion sensing controller. A user study comparing outstanding to regular teleport found a significant increase in spatial orientation, with no VR sickness or difference in presence.

A very similar approach was presented by Griffin et al. [19] which was published at the same time; called Out-of-body locomotion. A difference with outstanding is that in 3PP the users can steer their avatar using the touchpad on their controller which offers more precise navigation flexibility. If the user breaks line of sight with their avatar it automatically switches back to 1PP. A user study compares out-of-body locomotion to regular teleportation using an obstacle navigation task and found that it required significantly fewer viewpoint transitions with no difference in performance or VR sickness incidence.

3PP-R [13] brings 3PP to VR by rendering a miniature world that orbits with the user's viewpoint and which shows a miniature avatar. The avatar's ability to mimic the user's motions is limited because only the hands and HMD are tracked. Users navigate their avatar primarily using a controller and though positional tracking is supported this is bounded by tracking space boundaries. Performing a user study, the authors found 3PP-R to lead to less VR sickness than 1PP.

## 3 DESIGN OF EMBODIED 3PP LOCOMOTION

Prior studies [18,32] show that a 3PP can offer high embodiment but it requires visuo-motor synchronicity between the user and avatar. This is generally facilitated using full-body motion capturing which

either requires extensive user instrumentation [10, 15, 18] and (or) an array of (depth) cameras [32]. For locomotion, full-body tracking facilitates real walking which offers high presence [18] but this is generally bounded by available tracking space (i.e., when using external cameras) or available physical space (i.e., when using wearable sensors).

On most consumer VR systems, users rely on a combination of real walking and an alternative locomotion technique (ALT) like teleportation. Though teleportation allows for navigating at scale, it offers a low presence [8] which is a problem for when using it for 3PP locomotion as it can be assumed that this would offer a low embodiment. Other hybrid locomotion techniques have been proposed that aim to offer high presence by combining real walking with an ALT, e.g., walking-in-place (WIP) [6], arm swinging [31] or head-tilt [51] that offer high presence because they generate some of the proprioceptive/vestibular cues that are generated during real walking.

To facilitate 3PP locomotion with a high sense of embodiment, we propose a hybrid locomotion method that combines full body tracking based real-walking with head-tilt input. Head-tilt is a subset of leaning input (i.e., whole body tilt), a type of input that has been popularized by hover-boards [51]. Leaning has been explored for virtual locomotion where it has been found to offer high presence [30, 55]. The choice for head-tilt as opposed to other high presence ALTs like WIP or arm swinging was motivated by positive results from earlier studies. For an obstacle navigation task, head-tilt outperformed both WIP and a controller, while there was no significant difference in presence compared to WIP. There was also no significant difference in VR sickness compared to a controller [51]. For a bimanual target acquisition/deflection task, head-tilt offered a significantly higher presence than using a controller or teleportation though its performance was lower than using teleport [20]. No significant difference in VR sickness was found for head-tilt, WIP and using a controller. An earlier study also found no significant difference in VR sickness incidence between leaning input and a controller [30]. We have not found any research that found head-tilt input to increase VR sickness, and there is evidence that head-tilt can reduce car sickness [54], which is closely related to VR sickness. From an implementation perspective, head-tilt can be implemented with a high accuracy using an inertial measurement unit (IMU) that is present in the HMD. Though WIP can be implemented with an IMU [52], arm swinging either requires using controllers [31] or skeletal tracking which are more likely to be less accurate. Head-tilt also allows the user to retain independent control of their hands. This is useful, for example, when grabbing or punching an object or enemy while running. A limitation of using head-tilt is that it impedes the user's ability to freely look around while locomoting [52].

WIP and arm swinging generally do not support omnidirectional navigation (e.g., moving laterally or backwards). For WIP this can be achieved in a handsfree way when combining WIP with head-tilt [51]. A closely related 3PP implementation [35] integrates WIP with positional tracking but this setup does not allow for omnidirectional navigation and requires users to keep facing the camera. Full-body motion tracking generally requires using expensive cameras or extensive user instrumentation. Consumer depth cameras are low cost and require no user instrumentation but to allow for occlusion free skeletal tracking from all directions, multiple cameras from different viewpoints [32] must be integrated. Some rudimentary skeletal tracking is possible using wearable sensors (i.e., VIVE trackers) but since they only track torso and feet this doesn't allow for accurately animating a 3D avatar (e.g., elbows & knee joints are missing).

Given these hardware and tracking considerations, we decided to implement our 3PP locomotion method to only require use of a single depth camera. To enable this we were inspired by a now abandoned 3PP control system popularly known as *tank control*

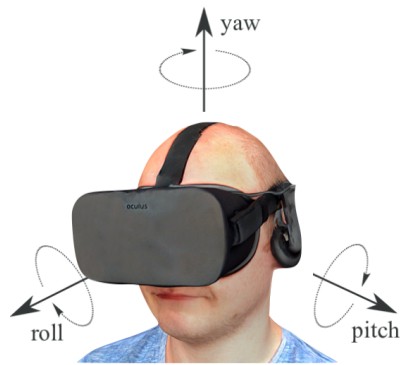

Figure 2: Head-tilt implemented using orientations acquired from an IMU.

in which the user controls their avatar movement relative to the coordinate system of the avatar. Up/Down input moves the avatar forwards/backwards in the direction it was currently facing while left/right rotates the avatar (clockwise/counterclockwise) This differs from current 3PP control schemes where movement is defined relative to the virtual camera's look direction. Early 3D games used 3D objects in combination with 2D pre-rendered backgrounds, which looked better but required using multiple predefined fixed in-game camera perspectives. This introduced a usability problem because if the user was navigating their avatar in a particular direction, cutting to a different camera perspective could make the game interpret user input differently from what was intended. Tank control didn't have this problem as it interprets movement relative to the avatar independent of a camera. When 3D rendering capabilities improved this control scheme was largely abandoned [38].

In a 3PP, the virtual camera is placed behind the avatar facing the avatar's back. Movement is issued relative to the virtual camera's look direction. Where in non-VR 3PP experiences the camera is typically controlled using a mouse or joystick, similar to most VR experiences, we pair the virtual camera to the position and orientation of the user's head to reduce visual-vestibular conflict. We then implement tank controls based on head-tilt for locomotion for the following reason. For optimal full-body tracking users must keep facing the camera to avoid occlusion. Tilt-based tank control ensures the user will always face the camera during locomotion as when users steer they rotate the virtual world relative to their avatar. When a user turns their head, the avatar will move out of the user's field of view as the avatar's location doesn't change. Our tank control scheme further ensures users will always be facing the camera by imposing a constraint that the user must be looking at their avatar to engage in locomotion.

If we used camera based movement, when the avatar rotates, we would have to rotate the camera as well to maintain an over the shoulder 3PP view. The optical flow from the camera orbiting around the avatar would be higher than when using tank controls and might induce VR sickness.

We combine tracking data from three different sources to implement our technique:

- **Positional tracking** data from the VR HMD is used to place and orient the virtual camera. The user can freely move around in the available tracking space and a grid is shown to indicate to the user that they are approaching the tracking boundary.

- **A depth camera** estimates full-body joint positions which are used to animate the virtual avatar in real time. If the user walks around in the available tracking space of the depth camera, the virtual avatar will do the same. For this to work in conjunction

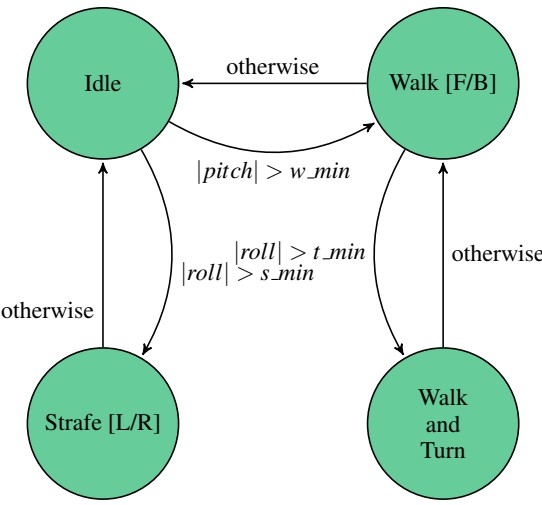

Figure 3: A finite state diagram of the Tilt Locomotion. $w\_min$, $s\_min$ and $t\_min$ represents the walking, strafing and turning thresholds respectively with the assumption that the user is facing the virtual avatar.

with the positional tracking input of the VR system, the two tracked spaces must overlap. The user also must adhere to the tracking space constraints. But since they are tethered to their avatar there is a risk of pushing or dragging the avatar outside the tracking space. We prevent this from happening by not letting the avatar cross the tracking boundary, which acts as a warning system to the user. For example, if a user is walking forward towards the camera, the avatar will stop when it reaches the tracking boundary and users must take care to return inside.

- **Inertial sensing**, acquired using the HMD's IMU is used to enable head-tilt locomotion. Head-tilt is calculated from the three possible degrees of rotation of the HMD; pitch, roll and yaw (see Figure 2). The pitch of the head dictates forward or backward movement, while the roll of the head is used for strafing when the user is not moving. We combine pitch and roll to implement tank controls when the user is moving forward. To allow users to be able to look around freely without engaging in movement, a dead zone has been defined and roll or yaw must exceed a certain threshold to activate movement. Roll and pitch can further be coupled to the avatar's locomotion speed to support variable locomotion speeds. A known limitation of head-tilt based locomotion [51] is that it limits the user's ability to freely look around as this changes the direction of locomotion. Users can freely look around when standing still while not looking at their avatar. During locomotion users can still look around with their eyes without moving their head, though this is constrained by the limited field-of-view of VR HMDs.

Figure 3 depicts a finite state machine diagram of the supported movement types. First, when standing still, as long as the user's head roll and pitch remain below their thresholds, the user can just look around freely and even turn their head 180° to look behind them. If the head tilt goes beyond threshold, the following can happen. If the user tilts their head forwards or backwards and passes the threshold, their avatar will walk forward or backward. If the avatar is standing still, tilting left or right will make the avatar strafe left or right. On the other hand, when the user tilts forward and then tilts left or right they will steer their avatar and the virtual world will be rotated around the avatar.

Tilt based locomotion implementation seamlessly integrates with positional tracking using the depth camera. Taking tracking constraints into account the avatar will be moved by any amount of observed skeletal displacement while a fixed distance between the camera and the avatar is maintained (except when approaching the tracking boundaries). Figure 4 provides an overview of all possible forms of locomotion and required corresponding head tilt inputs.

Additionally complex types of motion can be supported like jumping or crouching which can be activated using a gesture. For example, a short hop detected using inertial sensing can be used to trigger a much higher jump. A study has shown that switching between 1PP and 3PP still enables a high embodiment [15] and we believe transitioning from real walking to using head-tilt for locomotion is a smaller change than a 1PP to 3PP transition and will preserve a high embodiment.

## 4 USER STUDY

The goal of the user study was to evaluate the performance, usability, sense of embodiment and VR sickness incidence of our embodied locomotion method and to compare it to using a controller.

### 4.1 Instrumentation

Full-body skeletal tracking was accomplished using a Microsoft Azure Kinect DK. For our experiment, it operated with a resolution of 640x576 pixels at 30 frames per second. Latency was measured at 35ms which we deemed acceptable [2]. We placed this camera at a height of 1m on a tripod stand, which based on preliminary trials seemed to be most optimal for skeletal tracking with a user located at around 2m distance. From our experience, we found the Kinect sensor to prefer certain distances depending on the user's height and body type. That's why, before conducting the experiment, we made sure that the Kinect sensor was able to properly track each user at the distance they were standing and made necessary adjustments if needed.

For our HMD, we used the Oculus Rift S, a popular PC VR platform that allows full inside-out tracking of the HMD and two controllers using multiple cameras housed in the headset. The Oculus Rift S was specifically chosen because of its inside out tracking capability. Because the Kinect sensor projects infrared (IR) dots, VR systems that also rely on tracking using infrared light can cause interference. Specifically, we tested the Vive Pro and it wasn't compatible with our setup.

The Oculus Rift S offers a 1440x1280 per-eye resolution at 80 Hz and a variable field of view of around 110°. We used a High end PC (Ryzen 7 1700X, 16GB RAM, NVIDIA GTX 1080Ti) to run our VR application. For our study, we configured our tracking space to have a size of 2.5m x 2.5m, which is an average sized tracking space [3]. The Kinect camera was configured to operate within a depth range of approximately 3 meters(0.5 - 3.86m). The 75° field of view of the camera means that the width of the tracked space decreases as we move closer to the camera. As a result the tracked width was slightly lower compared to the VR system near the camera. The two tracking spaces were aligned to have maximum overlap. SteamVR's chaperone system keeps the user within the available tracking space and thus also keeps them visible to the Kinect camera most of the time.

For our study, since we are assessing locomotion performance, we compare our technique to using a Microsoft Xbox one wireless gamepad. Though trackpad or thumbstick input is available on VR motion sensing controllers, most commercially successful 3PP VR experiences (e.g., Lucky's Tale) are primarily experienced seated using a gamepad and participants are also most likely to be more familiar with a gamepad.

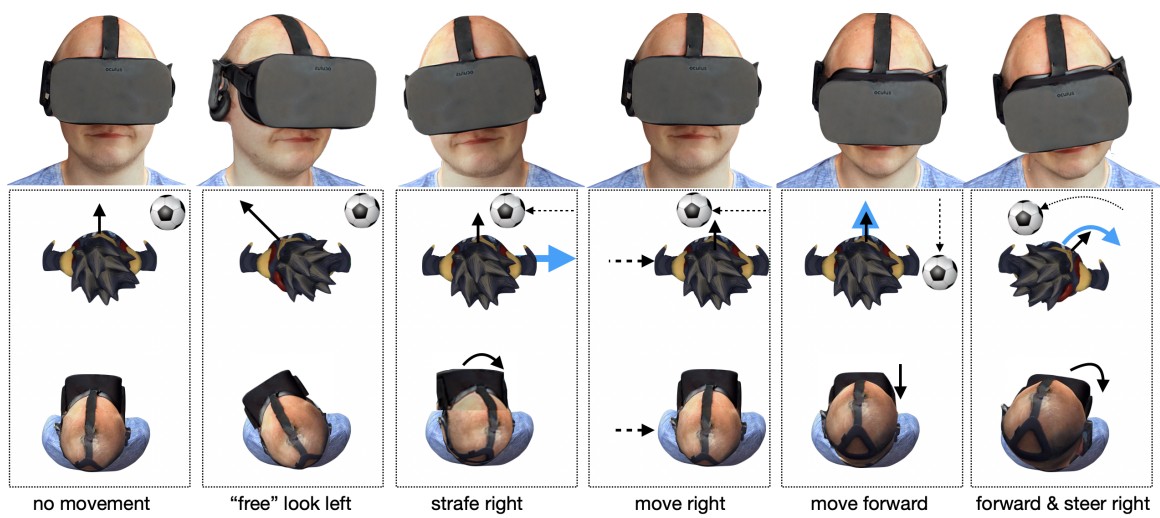

| no movement | "free" look left | strafe right | move right | move forward | forward & steer right |

Figure 4: Examples of how particular head-tilt motion is interpreted into avatar locomotion.

## 4.2 Virtual Environment

For our navigation task, we designed a virtual environment with a path for the user's avatar to follow. Path based navigation tasks have been used in closely related studies on VR sickness [4, 14]. We designed a winding path in an open environment that was demarcated by wooden boards (see Figure 5. The path contains a few sharp angles and turns requiring fast and precise controls. It has been a criticism [19] of existing studies, that most locomotion methods are evaluated in use cases that only involve navigation and not interaction with the environment. Though this seems to be a quite common use case for many VR experiences like games.

Since we were interested in evaluating the embodiment of our 3PP locomotion method, we designed an obstacle course that requires navigation but also interaction with objects. We made sure that it was long enough to take at least 7-10 minutes of time to run from start to finish. A study on VR sickness found that 2 minutes of optical flow exposure using a VR HMD [47] is already enough to elicit VR sickness symptoms in participants susceptible to it.

We placed 22 obstacles in the form of log stacks on the path. Users were tasked with jumping over these obstacles. We also put 136 balloons along the path with at least 5 meters between each balloon. 68 balloons were placed to the left and 68 to the right of the center of the path to compensate for handedness. We asked the users

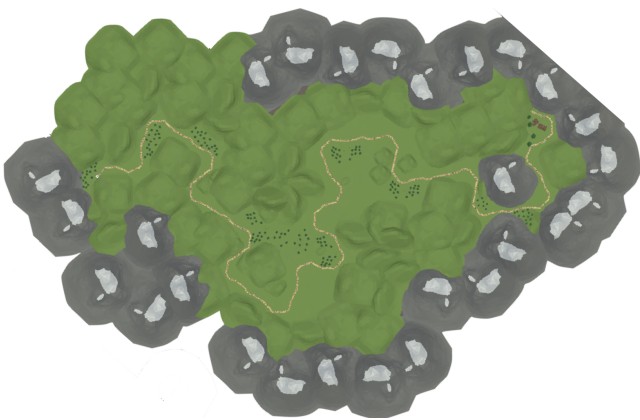

Figure 5: Virtual environment showing the path to be navigated.

to pop the balloons by hitting them using the avatar's hands and balloons only pop when there is a collision with the hands. Figure 6 shows both tasks in the virtual environment.

We developed the environment in Unity 2019.1.11f1. SteamVR plugin version v2.5 was used to implement the VR functionality. A $\delta$ of 1.0 was used so a 1.0 meter displacement in the real world corresponded to a 1.0 meter viewpoint translation in the virtual environment. We used the 3D avatar that came with an Azure Kinect example package for Unity [1]. To follow the avatar from a 3PP, we implemented a follow camera. A point at a height of 1.8m and at distance of 1.65m behind the avatar was selected as the target for the follow camera based on preliminary trials. The camera is always trying to reach this target location smoothly using a sigmoid function which helps dampen out motions from the user's avatar jumping. The camera is also rotating smoothly with the goal of matching the avatar's forward direction. We implemented the two locomotion methods in 3PP.

**Embodied locomotion.** As described in the design section this method combines outputs from the HMD positional tracking, and IMU and Azure Kinect sensor. Roll and pitch are interpreted to support navigation in any of the four egocentric directions that are easy to interpret by the user as this maps to joystick controls. Navigation by means of head tilt is enabled only when the user is facing the avatar. Whether the user is facing the avatar is detected by calculating the angle between the HMD's forward vector and the vector towards the avatar from the HMD's position. If this angle is below a threshold (which in our study was set to 15°), the user is considered to be facing the avatar. Also, we only engage in movement when roll or pitch exceed a minimum threshold. This allows users a greater freedom to freely look around.

Here, we further explain Figure 3 in terms of our implementation. The user starts in the idle state. Given they are facing the avatar, the users then have the option to make the avatar walk forward/backward or strafe sideways. Walking forward is enabled by a forward head tilt above threshold (threshold, w_min = 14°). When walking backwards, w_min was set to -11°. While the avatar is walking, if the user rolls their head to the left or right, the avatar will turn in the left or right direction respectively (threshold, t_min = 20°). On the other hand, if the avatar is standing still, the head roll will make the avatar strafe to the left or right (threshold, s_min = 20°). Values for these thresholds were determined experimentally from a small number of preliminary trials. Each of the threshold values w_min, t_min and s_min is accompanied by a maximum value which are

w_max (15°when walking forward and -11°when going backwards), t_max (28°) and s_max (30°) respectively These values were also determined experimentally. We use these minimum and maximum values with an inverse linear interpolation function to get our final input values in the range of 0 to 1. These input values are used to linearly interpolate between movement animations. We used the 'root motion' feature of our movement animations. This means that the avatar locomotion speed is coupled with the particular animation being played and it's speed. In our implementation the locomotion speed ranges between 0 and 4.5 m/s.

We implement jumping by calculating the headset's speed in the global up direction and comparing it against a predefined threshold. We maintain a moving average (n=4) of the speed to smooth out the data and avoid accidental jump commands. While the avatar is in air, the head tilt can be used to manipulate how far and in what direction the avatar jumps to an extent. This is done by applying a physics force to the avatar that we scale using the tilt input.

The Kinect sensor is used for skeletal tracking of the body. We map the joint orientation and position data to the avatar. Thus the movements of the user and the avatar are coupled. The Kinect is capable of estimating 32 body joint positions. To implement body tracking functionality in Unity we used the Azure Kinect Example Project asset as the basis. This package, however, had to be modified to achieve our desired functionality, e.g., masking part of the body to be controlled by body tracking while other parts by animation. The tracked skeleton by Kinect can show signs of jitters in the joints. To mitigate such anomalies, the example project comes with a 'smooth factor' option that linearly interpolates between previous joint positions to create a more stable skeleton. We set this to a value of 10 for our study.

When the user is locomoting using head-tilt, we animate the legs using a default animation clip and only the upper body will match any motions made by the user. This breaks visuo-motor synchronicity which could be detrimental to embodiment. However, with no animation it looked like the avatar was flying while dragging their feet which in preliminary trials seems to induce a lower sense of embodiment. Users can still move their avatar arms while walking forward and interact with objects etc. When not locomoting there is full body visuo-motor synchronicity and users can walk around as long as they remain visible to the sensor.

**Controller based locomotion.** This uses a standard 3PP control scheme where the left analog stick of the controller is used for avatar movement. Instead of using the right analog stick for rotating the camera as is common in non-VR 3D experiences, the users' HMD controls the camera which minimizes visual-vestibular conflict. The left and right bumpers of the controller are used to activate the left hand and right hand punch respectively. To activate jumping, we used one of the buttons (A) on the controller. In this control scheme, the avatar locomotion speed and animations work the same way as the Embodied scheme. The difference is that instead of head tilt determining the input, we instead use the left analog stick's input. Pushing the stick left-right is analogous to head roll while pushing it forward-backward is comparable to head tilt.

### 4.3 Experiment Design

The experiment was a 1 X 2 design with locomotion method as the independent variable (two levels: embodied and controller). We inspect the effect of this factor on task performance, usability, embodiment and VR sickness. To account for order effects, half of the participants started with the embodied condition (Group A) while the remaining half started with the controller condition (Group B) to compensate for any learning effects. Because the effects of VR sickness can linger for up to 24 hours; to minimize the transfer of VR sickness symptoms across sessions, we conducted each session on a separate day with at least 24 hours of rest between sessions.

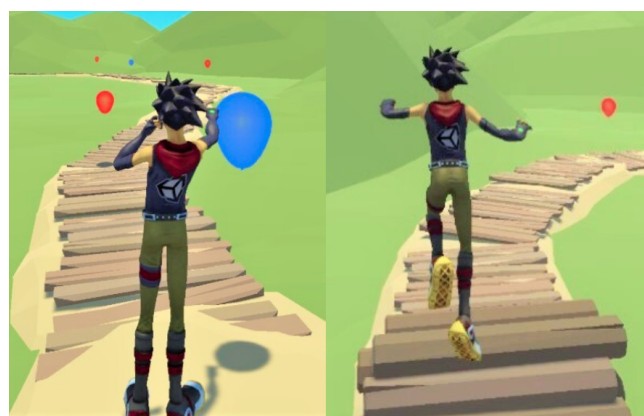

Figure 6: Tasks that users were required to perform during the navigation task. Left, punch a balloon, Right, jump over an obstacle.

### 4.4 Procedure and Data Collection

The experiment was conducted in a user study space that was free of noises and physical obstacles. When participants arrived for the first session they were briefed on the goal of the study, the outline of the experiment, the risks involved, the data collected, and the details of the training and experiment sessions. The distance between the VR HMD's lenses was adjusted to match the participant's interpupillary distance (IPD). Participants were then asked to stand in the middle of the tracking space. We first made sure that the Kinect sensor could track each participant properly. Then the users were assisted with putting on the VR headset so that they could start the training session.

The goal of the training session was to familiarize the participant with the controls used for the traditional 3PP control scheme and the techniques used for the embodied 3PP locomotion method. Participants were given an opportunity to try out both locomotion techniques in a short task that was similar to the experiment task.

Upon completing the training task, participants started the first experiment session where they were instructed to follow the obstacle course at their own pace. During the experiment session we recorded the number of balloons popped, number of obstacles jumped over, total duration of the experiment session and the amount of time spent walking on the path.

After completing each experiment session, participants were asked to fill out three questionnaires: 1) a Simulator Sickness Questionnaire (SSQ) [25] which is a standardized questionnaire used to measure the incidence of VR sickness, 2) a usability questionnaire which allowed participants to provide qualitative feedback about usability of the technique they just experienced, and 3) a standardized avatar embodiment questionnaire [16] to measure the embodiment of the avatar. As recommended by the authors of the SSQ [24] we use it only to assess post exposure VR sickness symptoms.

In this study, we use the standardized avatar embodiment questionnaire to address three aspects of virtual embodiment that are applicable to our experiment: body ownership, agency and motor control, and location of the body [16]. This questionnaire was developed for assessing embodiment in 1PP, while here we try to adopt it for avatar embodiment in 3PP which is different. Thus, following the recommendations of the standardized questionnaire, we only use a subset of the questions (e.g., Q1 to Q14) that are needed to calculate the metrics for these three aspects given whether we thought the questions were relevant to 3PP and the navigation task we had participants perform. The responses to the individual questions are combined into these three metrics based on the formulae provided in [16].

| | 1 | 2 | 3 | 4 | 5 | |
|---|---|---|---|---|---|---|
| Never | | | | | | Very frequently |
| **How often do you play video games of any kind?** | | | | | | |
| | 7.1% (1) | 28.6% (4) | 28.6% (4) | 21.4% (3) | 14.3% (2) | |
| **How familiar are you with 3P navigation using a controller?** | | | | | | |
| | 0.0% (0) | 14.3% (2) | 21.4% (3) | 35.7% (5) | 28.6% (4) | |
| **How frequently do you use VR?** | | | | | | |
| | 42.9% (6) | 28.6% (4) | 14.3% (2) | 7.1% (1) | 7.1% (1) | |
| **How frequently do you get VR/motion sick?** | | | | | | |
| | 42.9% (6) | 21.4% (3) | 35.7% (5) | 0.0% (0) | 0.0% (0) | |

Figure 7: Summary of participants ratings of their frequency of playing video games, familiarity with 3P navigation using a controller, frequency of using VR and their tendency of getting motion or VR sick on a scale of 1 (never) to 5 (very frequently). The results are reported in the form of *percentage (count)*.

Finally, after completing both experiment sessions, participants were asked to fill out a post-study questionnaire which was used to collect demographic information that included their age and sex; and their frequency of playing video games, familiarity with controller based third person navigation, frequency of using VR, and tendency of being motion and/or VR sick using a five-point Likert scale. On average, the whole study took about 45 minutes to complete in two sessions. All participants were compensated with a \$15 Amazon gift card for their time, and the user study was approved by an IRB.

### 4.5 Participants

Recruitment of participants was significantly impeded by the Covid-19 pandemic and recruitment for our user study was shutdown halfway through. Nevertheless we were able to recruit fifteen participants for our study. One participant could not complete the study because she could not be properly tracked by the Kinect sensor, which was likely caused by her clothing (loose fitting dress). A total of fourteen participants (4 females, 10 males, average age=24.9, SD=4.6) were able to complete both sessions and their data is analyzed in this study.

### 5 RESULTS

Participants were asked to rate their frequency of playing video games, familiarity with controller based third person navigation, frequency of using VR, and tendency of being motion and/or VR sick on a scale of 1 (never) to 5 (very frequently). The results are summarized in Table 7. To measure task performance, we logged the position of the avatar in the virtual environment, time stamps, number of balloons popped, number of obstacles jumped and the percentage of time participants spent on the path (e.g., if participants didn't deviate from the path this number would be 100%).

We analyzed these quantitative results using a one way repeated measures MANOVA. For qualitative results, all participants answered an avatar embodiment questionnaire, an SSQ and a usability questionnaire after each trial. The responses collected through the embodiment and usability questionnaires were analyzed using non-parametric methods (Wilcoxon signed rank paired-test).

### 5.1 Task Performance

Table 1 lists the task performance results for both methods. For our analysis we used (1) total time, (2) % of targets hit, (3) % obstacles jumped, and (4) % of time spent on the track. A one-way repeated measures MANOVA found a statistically significant difference between locomotion techniques on the linear combination of the dependent variables ($F_{4,10} = 3.689$, $p = .043$, Wilk's $\lambda =$

| Locomotion type | **Embodied** (SD) | **Controller** (SD) |
|---|---|---|
| Total time (s) | 519.47 (104.3 ) | 423.25 (6.6 ) |
| % targets hit | 89.86 (6.4 ) | 95.75 (3.7 ) |
| % obstacles jumped | 92.53 (1.1 ) | 99.35 (1.7 ) |
| % on track | 95.60 (3.8 ) | 97.59 (.5 ) |

Table 1: Quantitative results for each locomotion method. Standard deviation listed between parentheses.

.404, partial $\varepsilon^2 = .596$). Mauchly's test of sphericity indicated that the assumption of sphericity had been met.

Follow up univariate tests found statistically significant differences between locomotion methods for total time transitions ($F_{1,13} = 12.710$, $p = .003$, partial $\varepsilon^2 = .494$), targets hit ($F_{1,13} = 11.910$, $p = .004$, partial $\varepsilon^2 = .478$), obstacles jumped ($F_{1,13} = 5.571$, $p = .035$, partial $\varepsilon^2 = .300$). However, there was no statistically significant difference between locomotion methods for time spent on track ($F_{1,13} = 3.828$, $p < .072$, partial $\varepsilon^2 = .227$).

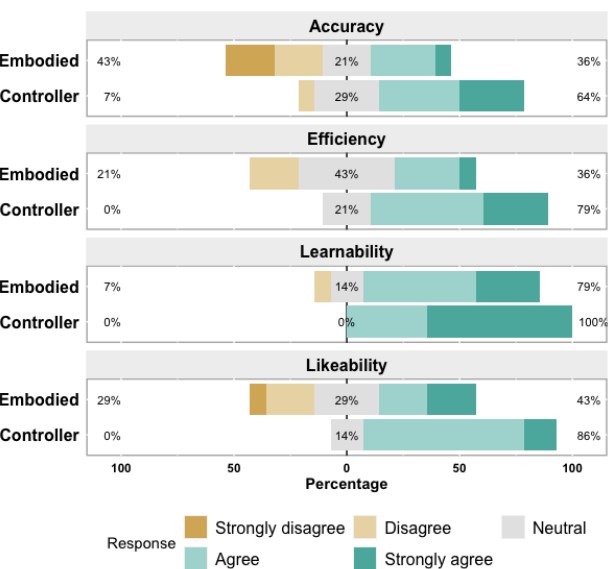

Figure 8: Diverging stacked bar chart of the percentages of the Likert scores for the subjective usability rankings of each locomotion method.

### 5.2 Usability

After completing each session participants were asked to rate the locomotion method they just tested in terms of accuracy, efficiency, learnability and likeability using a 5 point Likert scale ranking from 1: strongly disagree to 5: strongly agree. The results are summarized in Figure 8. A Wilcoxon Signed-Rank test was used to analyze for differences in Likert scores. We found statistically significant difference for accuracy ($Z = -2.341, p = .019$), efficiency ($Z = -2.46, p = .014$), learnability ($Z = -2.124, p = .034$), and likeability ($Z = -2.077, p = .038$).

### 5.3 Simulator Sickness

We used the SSQ results to calculate the SSQ subscores: total score, nausea, oculomotor and discomfort as described in [57]. A one-way repeated measures MANOVA did not find a statistically significant difference between locomotion techniques on the linear combination of the SSQ subscores ($F_{3,11} = 0.185$, $p = .360$, Wilk's $\lambda = .756$, partial $\eta^2 = .244$). Mauchly's test of sphericity indicated that the assumption of sphericity had been met.

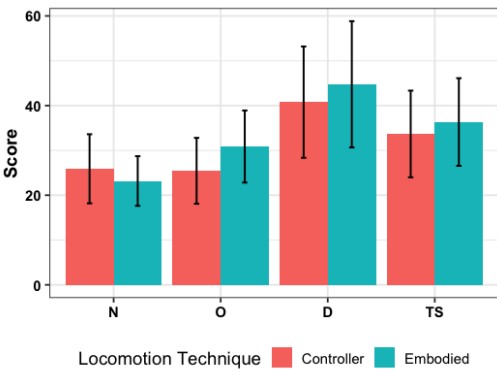

Figure 9: Summary (means) of the four subscores of the SSQ score: (N) ausea score (O) culomotor discomfort, (D) isorientation score and the (T)otal (S)everity score . Error bars show standard error of the mean.

## 5.4 Avatar Embodiment

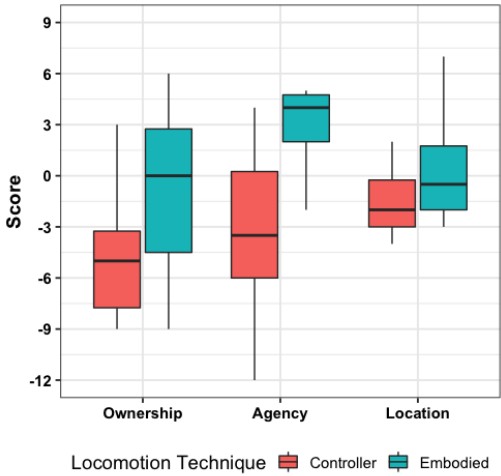

Figure 10: Embodiment scores for both methods measured by the metrics ownership of the body (Ownership), agency and motor control (Agency), and location of the body (Location).

Analyzing the avatar embodiment questionnaire responses, we found that participants preferred the embodied locomotion method over the controller method. The results are summarized in Figure 10. Participants reported a significantly higher ownership of the virtual avatar when using the embodied locomotion method ($Z = -2.482, p = .013$) than when using the controller based locomotion method. Participants also reported significantly higher scores of agency and motor control when using the embodied locomotion method ($Z = -2.485, p = .013$) compared to the controller based locomotion method. Additionally, as measured by the location of the body metric, participants thought the embodied locomotion method provided significantly higher embodiment illusion ($Z = -2.131, p = .033$) compared to the controller based locomotion method.

## 6 DISCUSSION AND FUTURE WORK

**Performance**. Not surprisingly, in terms of performance using a controller performed significantly better than our embodied locomotion method. Controllers require very little physical effort to be used and prior studies have repeatedly found that a controller is faster and easier to use mostly because most users are highly familiar with this type of input. The differences in performance seem quite reasonable, e.g., total time (22% slower), balloons hit (6% lower), obstacles jumped (7% lower) and time on track (2 % lower).

**Usability**. Using a controller was found to be more accurate, efficient, easier to use and overall preferred to our embodied locomotion method. The significantly higher familiarity with using a controller (as evident due to the 100% agree or higher score for learnability) largely explains why users found a controller more accurate, efficient and better liked than our embodied locomotion method. However, to contextualize these usability results it is important to distinguish locomotion from interaction (e.g., interacting with objects). To hit a balloon and jump over an obstacle our embodied locomotion method required real physical movements (i.e., punching and jumping), which is slower and more error prone than pressing a button. Though the Kinect tracking is pretty accurate there was a small amount of latency, especially affecting jumping-which would sometimes cause participants to run into the obstacle rather than them jumping over it. Hitting a balloon while running also required precise timing and was just harder to perform with our embodied method. Some participants were observed to navigate backwards when they missed hitting a balloon so they could try hitting it again, which added to their time. Looking at locomotion efficiency, there was no significant difference in percentage of time on track for both locomotion methods. Overall these factors contributed to a worse rated usability for our embodied method, which was further exacerbated by participants' high familiarity with using a controller. Though participants were given enough time to familiarize themselves with our embodied interface, over time with greater proficiency the rated performance and usability could increase.

**Embodiment**. Our study did find evidence that our embodied locomotion method offered a significantly higher avatar embodiment than when using a controller, which was the main objective of our method. The motivation to compare our method to a controller was made largely for benchmark purposes with no reasonable expectation that our embodied locomotion method would outperform a controller for performance or usability (users were more familiar with a controller). To enable embodiment illusion research [37], locomotion performance may not seem to be an important factor, but usability probably is. We did not explore using our 3PP locomotion method for embodiment illusion research, which is something we hope to explore in collaboration with experts in this area.

Though our approach lets users see their avatar, existing embodiment illusion research uses 1PP with a virtual mirror which allows for face-to-face interaction, which is important [17]. Because our approach uses a fixed camera from behind, you only see the avatar's back, but we aim to develop a hands free control scheme that lets users rotate their camera to allow users to see their avatar and their avatar's face from different angles. Our study did not assess presence as this is determined by many factors including the VR experience itself, but we hope to substantiate this in future work.

**VR Sickness**. There was no significant difference in VR sickness incidence as measured using the SSQ between both locomotion methods. Head-tilt generates some of the vestibular cues that are present in walking, which are notably absent when using a controller and so there was the possibility this could alleviate visual-vestibular conflict. However, we did not find any differences and this corroborates an earlier study that compared a controller to using head-tilt input for locomotion (using a 1PP) that also did not find a significant difference in VR sickness as measured using SSQ [51]. An important finding was that VR sickness incidence was low. Six participants out of the fourteen were asymptomatic, and the overall observed average total SSQ scores were very low (i.e., 33/36 out of a maximum of 235) which corresponds to very mild VR sickness. Though experimental conditions differ, many prior studies [14,28,33,59] have found that a controller generally induces moderate to high levels of VR sickness. A notable difference is that prior studies all used a 1PP where we

used 3PP. The Low VR sickness score could be because the users were looking at an avatar during locomotion, which might have served as a rest frame [40]. Because no studies have investigated how perspective affects VR sickness, this is something we certainly aim to investigate in future work.

**Limitations**. Our user study involved a low number of participants (n=15) as our University stopped human subject research campus wide halfway through our recruitment due to COVID-19. Our embodied locomotion method will only work with HMD's that feature non-IR inside-out tracking as this does not interfere with the IR used by the depth camera. Another limitation imposed by the specifications of the Kinect Azure camera that we used is that in order to be always visible, the tracking space must be defined within the depth range of the camera. Larger tracking spaces could possibly be supported using multiple Kinect cameras and if the user is always visible from every angle, tank controls can be abandoned. A related issue is that the camera doesn't track a rectangular region. So getting a perfect match between the tracking space of the VR system and the Kinect wasn't possible. Multiple cameras can solve this issue as well by covering a larger space than what the VR system does. We have been able to integrate positionally tracked controllers into our method, which improves skeletal tracking and provides rotational information for the hand joints, which is useful for example when holding an object. Our study only evaluated navigation and limited interaction with objects. Our study did not require participants to navigate using positional tracking input (e.g., real walking), though that certainly was possible. However, this increases the likelihood of users stepping outside of the tracking space where visuo-motor synchronicity between the user and avatar cannot be assured which is likely detrimental to embodiment.

# 7 CONCLUSION

We present a novel embodied 3PP locomotion method that blends real walking using full body skeletal tracking with head-tilt based locomotion. In addition to being able to fully see your avatar our locomotion method allows users to navigate beyond available tracking space constraints and is minimal in terms of required sensors (e.g., a single depth camera). Our user study found that controller input was better in terms of performance and usability, but we did not find any difference in VR sickness incidence. Our method offered a significantly higher avatar embodiment than using a controller, which is an important finding for games as well as embodiment illusion applications. Given the low VR sickness scores we measured when evaluating both of the third person locomotion interface, it suggests that perspective might play a role in VR sickness incidence.

# 8 ACKNOWLEDGEMENTS

This work is supported by NIH grant P20GM103650 and NSF grant 1911041.

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
