# OpenReview forum: "Embodied Third-Person Virtual Locomotion using a Single Depth Camera"
_graphicsinterface.org/Graphics_Interface/2021/Conference — GI 2021_

### Official Review · AnonReviewer2 · 2021-01-08
**Embodied Third-Person Virtual Locomotion using a Single Depth Camera**

**Rating:** 7
**Confidence:** 3

**Review:**

The authors present a third-person perspective locomotion technique based on the integration of skeletal tracking and head-tilt-based input. A user study shows that the proposed method enables high avatar embodiment when compared with a controller.

Comments:
- The paper is very well written, and the proposed method is clearly presented.

- One major issue is that I find that the virtual environment is straightforward. When comparing locomotion techniques, one has to take extra care when designing the task and virtual environment. In the paper, the task is navigating a path in an "open" virtual environment containing minimal visual detail. Having an open virtual environment where the target locations/path are visible from the user's current position can significantly and negatively affect the comparison between locomotion techniques. As stated in the paper, "a controller was found to be more accurate, efficient, easier to the user and overall preferred ...". This could, in fact, be attributed to the trivial task given the virtual environment used. A better way of conducting the user study would be to create a virtual environment where target locations are obscured from the user's view. This forces the user to maintain route directions in memory and update their mental map of their position by looking around for landmarks. Given the simplicity of the task and environment used, in my opinion, the comparison has reduced to choosing a preferred method due to familiarity, which, not surprisingly, is the controller.

- In the Introduction, there is a reference missing [?]. Similarly, in Section 3 there is another.

- Related work: I would suggest including, at the end of the section, a paragraph comparing how the proposed technique differs from the ones mentioned, and in particular with [31], which is using a similar setup. Why haven't you used multiple depth cameras like [31]? This would have solved the problem of the user stepping outside the tracking area. Also, the users wouldn't have to face the camera all the time to avoid occlusions.

- One participant could not complete the study because of tracking problems. Why was the tracking failing for this one participant?

---

### Official Review · AnonReviewer1 · 2021-01-11
**This study explores the third person perspective with first person perspective when using VR applications and how this impacts upon measures such as performance, usability, embodiment, VR sickness**

**Rating:** 8
**Confidence:** 4

**Review:**

This is an interesting study exploring the impact of the third person perspective compared with first person perspective when using VR applications. Whilst the research presented here is novel, I feel the introduction and background sections could have been somewhat simplified to more clearly outline the research problem that is being explored here. More diagrams could have been used to explain some of the visual concepts that have been attempted to be explained purely in text. Similar issues persist in the User Study section, for example, a diagrammatic representation of the experimental setup would have helped to show the reader what the system architecture/technical instrumentation setup is that was deployed in the experimental system. It is difficult to conceptualise how the system was set up purely from the textual descriptions provided. The sample size of this study is small, and it therefore difficult to generalise from the results presented here, however, the results are interesting, although not surprising. For example, the improved performance using the gamepad controller was to be expected (although looking at the sample, there may have been several participants that may not even be very familiar with using gamepad controllers), also in terms of usability, again the gamepad controller was more accurate and efficient (quite possibly in part due to familiarity?). Embodiment was found to be improved in the 3PP condition, again probably not surprising, but a valuable finding nonetheless. And third person perspective with first person perspective VR sickness was similar in both conditions - I'm not sure why the authors state that this suggests that perspective *could* play a role in VR sickness? This is suggested both in the abstract and the conclusion, but there is no explanation as to why the authors think this could be the case? Overall, the abstract and conclusions are almost identical and also weak, and do not do justice to the work that is presented in the main body of the paper, this could be improved somewhat. Overall a good study, unnecessarily complex in terms of the background and related work sections which could also be significantly improved via the use of appropriate diagrammatic representations instead of complicated textual descriptions/attempts an explanation. Overall, however, a good piece of work. It's a shame the sample size is so small, as the contributions had the potential of being greater if the sample was bigger.

---

### Official Review · AnonReviewer3 · 2021-01-13
**The paper presents the use of a single depth camera for skeletal tracking which is used to provide third person virtual locomotion controlled by head tilt and/or a hand held controller. User study is conducted for VR sickness, performance, usability and embodiment. It is a detailed study, though COVID 19 limited the number fo participants.**

**Rating:** 6
**Confidence:** 3

**Review:**

In VR first person  perspective is considered the better choice for embodiment. Third person perspective has its own advantages as evident in many 3D games.  This paper explores the third person perspective for VR. It uses a single depth camera for skeletal tracking which is then used for virtual locomotion. They conduct a well designed user study experiment to compare locomotion and interaction through head tilt as one option for locomotion and a controller as the other option. User study seems to indicate that embodiment is better with the head tilt as compared to use of the controller, whereas usability and performance are better with the controller. VR sickness was not significant.

Even though COVID 19 limited the number of participants, the study seems to have been well conducted. The overall implementation and experimental results are worth reporting.

---

### Meta-Review · Area_Chair1 · 2021-01-15

**Recommendation:** Accept
**Confidence:** 4

**Metareview:**

Overall this is a good study of third person perspective in VR locomotion and interaction. Some of the explanations could improve with diagrams.  Conclusions could be better written.  There are some typos and problems with citations which should also be corrected. All reviewers agree that the paper should be accepted. Please look into detailed comments from reviewers and modify the submission accordingly.

---

### Decision · Program_Chairs · 2021-01-16

Accept